# Decision Levels and Resolution for Low-Power Winner-Take-All Circuit [note 1]

**DOI:** 10.3390/s23146247

**Published:** 2023-07-08

**Authors:** Ruxandra L. Costea

**Affiliations:** Electrical Engineering Department, Electrical Engineering Faculty, Polytechnic University of Bucharest, 060042 Bucharest, Romania; ruxandra.costea@upb.ro

**Keywords:** Winner-Take-All, subthreshold, decision levels, resolution, mismatch

## Abstract

Sensors in many applications must select the largest element in a sequence of currents. This can be performed in an analog way by the Winner-Take-All (WTA) circuit. This paper considers the classic version of the WTA Lazzaro circuit, working with MOS devices in a subthreshold regime. Since the separation of the gainer by analytically computable “decision levels” has recently been introduced, this paper aims to numerically verify and discuss these levels and their dependence on circuit and device parameters. For $V_{T}$, the threshold voltage of MOS devices, which is primarily responsible for differences between components (mismatch), its relationship with the output voltages is theoretically demonstrated and numerically checked.

## 1. Introduction

Winner-Take-All (WTA) circuits are one of the most important building blocks in analog parallel signal processing, such as spatial acquisition and tracking, sound localization, image processing and neuromorphic systems [1,2,3,4,5,6,7]. The main function of a WTA is to select the highest input signal among multiple inputs, so two-input WTAs can also be used as half-wave and full-wave rectifiers [8,9,10]. Many WTA proposals can be found in the literature, such as voltage-mode configurations based on differential pair structures [11,12] or on inhibitory and local excitatory feedback loop circuits [13,14,15,16]. The first approach suffers from complexity and a high power comsumption, whereas the second approach has potential stability issues due to positive feedback.

In recent years, the processing of nanoamps has become increasingly important. These currents can come from sensors inside the human body—[17], from chemical reaction sensors, from motion tracking or from computer memory—[18]—to name just a few examples. In fact, the analog processing of very small signals was used in “neuromorphic” circuits initiated by Carver Mead at Caltech in the early 1990s [19,20]. In that context, the first W (inner) T(ake) A(ll) circuit appeared, known today as the “Lazzaro Circuit”—[21]. Its simplicity that leads to space savings on integrated chips has distinguished it technologically. Many improvements to the Lazzaro circuit have been proposed in the meantime [22,23], but the basic principle and configuration have not changed.

In [24], Sekerkiran et al. proposed a modified version of Lazzaro’s WTA, which improved the resolution without requiring positive feedback, thus avoiding major stability issues. Their approach consisted of using an aditional transistor per cell to increase the open loop gain and, therefore, improve resolution. However, both Lazzaro’s and Sekerkiran’s WTAs need a high voltage swing at the input nodes to turn on the winning cell, which results in a slow response to abrupt input current changes.

The parameters of MOS and their interconnections must be as identical as possible. Integrated circuit technology, engaged in a race to reduce the size of chips and circuits, cannot ensure the strict identity of the parameters on the same chip or on different chips. Thus appears the so-called “mismatch”—[22,23,24,25], whose size is a criterion for the performance of the chips. It is all the more important as the circuit works with lower currents. In this way, the study of the variation in the performances of the circuits with MOS transistors working in the subthreshold (or weak inversion) when the model parameters slide around the design value is decisive.

This paper considers the original Lazzaro circuit, working in the subthreshold as a selector of the maximum current rank. For the list of output voltages, a parallel paper co-authored by the present author—[26,27]—introduced a “higher decision level”. Above it, only the highest rank (winner) in the output list must be placed. Similarly, a “lower decision level” must be above the second-largest rank and under the higher level. Both levels were rigorously defined and used to introduce the resolution performance of the selector.

In fact, we need two notions of resolution. One for the input lists—the input resolution—and one for the selection result—the output resolution. To be sensitive and efficient, a WTA fed with “crowded” lists must select the output through a “wide” separation. Below, after introducing the circuit static model in Section 1, we present the theoretical questions about decision levels and about the resolution in Section 3 and Section 4. In Section 5, the monotonic dependence of the winner size on the threshold voltage $V_{T}$ of MOS devices is proven. Section 6 contains numerically computed examples. Analytically computed decision levels and resolutions are extensively checked and analyzed.

## 2. The Circuit Model

For the subthreshold regime—i.e., when $V_{GS}\leq V_{T}$ and $V_{DS}\geq 0$—we use the usual MOS model [28] with the usual notations:(1)$$\begin{array}{c} I_{DS}=I_{0}\left[\exp\left(-\frac{V_{S}}{V_{t}}\right)-\exp\left(-\frac{V_{D}}{V_{t}}\right)\right]\exp\left(k\frac{V_{G}}{V_{t}}\right) \end{array}$$

Then, the steady state of the Lazzaro WTA circuit in Figure 1 (with all devices in the subthreshold) can be obtained by $I_{j}=I_{T_{j}}$ and $I_{C}=\sum_{j=1}^{N}I_{T_{j}^{★}}$ where $I_{T_{j}}$ and $I_{T_{j}^{★}}$ are the $I_{DS}$ currents for $T_{j}$ and $T_{j}^{★}$, respectively. Thus, we will move on to the following:(2)$$\begin{array}{c} U_{j}=V_{t}\ln{\left[1-\frac{I_{j}}{I_{0}}\exp\left(-k\frac{V}{V_{t}}\right)\right]}^{-1},\;\;\;\;j\in \bar{1,N} \end{array}$$
(3)$$\begin{array}{c} I_{C}=G\left(V,I\right) \end{array}$$
where
(4)$$\begin{array}{c} \begin{array}{c} G\left(V,I\right)=I_{0}\left[\exp\left(-\frac{V}{V_{t}}\right)-\exp\left(-\frac{V_{DD}}{V_{t}}\right)\right]\times \\ \sum_{j=1}^{N}{\left[1-\frac{I_{j}}{I_{0}}\exp\left(-k\frac{V}{V_{t}}\right)\right]}^{-k} \end{array} \end{array}$$

In the above equations, the MOS parameters are $I_{0}$, *k* and $V_{T}$ while $I_{j}$, $I_{C}$ and $V_{DD}$ are outside constant sources. For an input list of currents $I=\left(I_{1},I_{2},\ldots ,I_{N}\right)$, Equation (Equation 3) provides the common potential *V* with which Equation (Equation 2) gives the output voltages $U=\left(U_{1},U_{2},\ldots ,U_{N}\right)$.

Obviously, we have to make sure that all transistors $T_{j}$ and $T_{j}^{★}$ work in the subthreshold, which means [26,29] *V* and $U_{j}$ must be restricted to
(5)$$\begin{array}{c} 0\leq V\leq \min\left\{V_{DD},V_{T}\right\} \end{array}$$
(6)$$\begin{array}{c} 0\leq U_{j}\leq V_{T}+V,\;\;\;j\in \bar{1,N} \end{array}$$

As we prove in [26], the following restrictions are sufficient:(7)$$\begin{array}{c} V_{T}<V_{DD} \end{array}$$
(8)$$\begin{array}{c} I_{0}\leq I_{M}\leq I_{0}\exp\left(\frac{kV_{T}}{V_{t}}\right) \end{array}$$
(9)$$\begin{array}{c} \frac{I_{0}}{N-1}\leq \Delta \leq \frac{I_{M}}{N-1} \end{array}$$
(10)$$\begin{array}{c} I_{C}\geq G\left(V_{T},\hat{\hat{C}}\right) \end{array}$$
where $I_{M}$ is the absolute maximum of currents allowed for processing. For Δ and $\hat{\hat{C}}$, see below. Let us note that the right side in (Equation 9) is not a restriction.

Finally, let us mention that, in [29], for the dynamic model of our circuit, the invariance of the solution in a weak inversion region as well as its asymptotic stability have been studied.

## 3. Decision Levels

To explain the issue of decision levels, let us start with a simple example.

Let us consider our WTA in the particular case of N=3, fed with the infinite number of lists in $L\left(3,I_{M},\Delta \right)$, that is, lists with three currents, no bigger that $I_{M}$ and separated from each other by the minimum distance Δ. The first list $I=\left(I_{1},I_{2},I_{3}\right)$ with the (decreasing) order $\sigma =\left(3,1,2\right)$ arrives at the WTA input—see Figure 2. The goal is to signal the “winning” rank σ1=3 of the largest current $I_{3}$, even in the extreme case when “the loser”—which is the second-largest current $I_{1}$—is at the minimum distance Δ, $I_{3}-I_{1}=\Delta$ and Δ is so small that the two are not distinguishable on the $\left[0,I_{M}\right]$ scale.

The WTA circuit translates the reading of the winner rank to the output list of voltages $U=\left(U_{1},U_{2},U_{3}\right)$, which has the same order $\sigma =\left(3,1,2\right)$. However, the winner $U_{3}$ is now split from the loser $U_{1}$ by a gap $\bar{D}-\underline{D}$, which is sufficiently large on the $\left[0,U_{M}\right]$ scale.

In fact, we have to have $U_{3}\geq \bar{D}>\underline{D}\geq U_{1}>U_{2}$. $\bar{D}$ is called “the upper decision level” and has the property that it is surpassed only by the winner. Thus, the outputs $\left(U_{1},U_{2},U_{3}\right)$ are compared with $\bar{D}$—see Figure 2—and rank 3 will be the unique winner.

Furthermore, $\underline{D}$ is called “the lower decision level”, and all the “losers” ($U_{1}$ and $U_{2}$ here) are under it. The distance $\bar{D}-\underline{D}$ is significant on the scale $\left[0,U_{M}\right]$, where $U_{M}$ is the maximum voltage. Returning to Figure 2, let us consider a second list $\left(J_{1},J_{2},J_{3}\right)$ from $L\left(3,I_{M},\Delta \right)$ applied at the input. Suppose that $J_{2}>J_{3}>J_{1}$ and the winner rank “2” has to be signaled. This is performed by obtaining the output voltages $\left(W_{1},W_{2},W_{3}\right)$ arranged as $W_{2}\geq \bar{D}>\underline{D}\geq W_{3}\geq W_{1}$, where the only rank surpassing the upper decision level $\bar{D}$ is “2”, the winner. The losers are below the lower decision level $\underline{D}$. The processing should be similar for any list from $L\left(3,I_{M},\Delta \right)$ when using the same decision levels $\bar{D}$ and $\underline{D}$ and the same circuit parameters.

For the input list $I=\left(I_{1},I_{2},\ldots ,I_{N}\right)$ written in the terminal order, let us denote $\sigma =\left(\sigma 1,\sigma 2,\ldots ,\sigma N\right)$ a permutation of the indices such that the currents in $I^{\sigma }=\left(I_{\sigma 1},I_{\sigma 2},\ldots ,I_{\sigma N}\right)$ are in decreasing order:(11)$$\begin{array}{c} I_{M}\geq I_{\sigma 1}>I_{\sigma 2}>\ldots >I_{\sigma N}\geq 0 \end{array}$$

Then, from (Equation 2), it is clear that the output voltages $U=\left(U_{1},U_{2},\ldots ,U_{N}\right)$ are in the same decreasing order, i.e.,
(12)$$\begin{array}{c} U_{M}\geq U_{\sigma 1}>U_{\sigma 2}>\ldots >U_{\sigma N}\geq 0 \end{array}$$ Both $I_{\sigma 1}$ and $U_{\sigma 1}$ are called “winner” while both $I_{\sigma 2}$ and $U_{\sigma 2}$ are called “loser”. To $U_{\sigma 3},U_{\sigma 4},\ldots ,U_{\sigma N}$ and to $I_{\sigma 3},I_{\sigma 4},\ldots ,I_{\sigma N}$ as well, we use the same name, “losers”.

We will assume that the input currents $I\in {ℜ}^{N}$ belong to the class $L\left(N,I_{M},\Delta \right)$, i.e., their components are inside the $\left[0,I_{M}\right]$ interval and are mutually separated by distance Δ at least. To abbreviate the writing, from here on, we will denote this class simply by L. Thus,
(13)$$\begin{array}{c} I_{\sigma j}-I_{\sigma \left(j+1\right)}\geq \Delta ,\;\;\;j\in \bar{1,N-1} \end{array}$$

This leads to the existence of $\left[C_{jm},C_{jM}\right]$ intervals in which each $I_{\sigma j}$ current is forced to belong:(14)$$\begin{array}{c} I_{\sigma j}\in \left[C_{jm},C_{jM}\right] \end{array}$$

Here, for each $j\in \bar{1,N}$,
(15)$$\begin{array}{c} C_{jm}=\left(N-j\right)\Delta \end{array}$$
and
(16)$$\begin{array}{c} C_{jM}=I_{M}-\left(j-1\right)\Delta \end{array}$$

We put
(17)$$\begin{array}{c} \hat{\hat{C}}=\left(C_{1M},C_{2M},C_{3M},\ldots ,C_{NM}\right) \end{array}$$
the list of maximum currents of each rank from (Equation 10). Note that the intervals in (Equation 14) do not overlap. All possible lists at the input (i.e., satisfying (Equation 7)–(Equation 10)) have in common the number of elements *N*, the maximum current $I_{M}$ and a measure of the agglomeration of the currents Δ. Let us denote by L this (infinite) family of input lists.

The raison d’$\hat{e}$tre of the WTA circuit is to identify the rank σ1 of the highest current $I_{\sigma 1}$ in list *I* and to achieve this for any input list in L without changing the parameters or configuration.

A recent co-work by the author of this paper—[26]—has introduced two “decision levels”, $\bar{D}$ and $\underline{D}$, which split the output list as follows:(18)$$\begin{array}{c} U_{M}\geq U_{\sigma 1}\geq \bar{D}>\underline{D}\geq U_{\sigma 2}>U_{\sigma 3}>\ldots >U_{\sigma N} \end{array}$$

Here, $\bar{D}$ is defined as the smallest winner of the output lists when all inputs in L are applied. Similarly, D is the highest loser for all inputs in L. The main attraction of these particular levels consists of the fact that they can be obtained “semi-analytically”.

Thus, it is proven that
(19)$$\begin{array}{c} \bar{D}=U_{1}\left(\bar{C}\right) \end{array}$$
where
(20)$$\begin{array}{c} \bar{C}=\left(C_{1m},C_{2m},C_{3m},\ldots ,C_{Nm}\right) \end{array}$$
is the L-list with currents in (Equation 15). This means that the upper decision level $\bar{D}$ is exactly the winner of the output list $U=U\left(\bar{C}\right)=\left(U_{1}\left(\bar{C}\right),U_{2}\left(\bar{C}\right),\ldots ,U_{N}\left(\bar{C}\right)\right)$ when the currents in $\bar{C}$ are the input. It is shown that they are the smallest possible currents of each rank, computable as in (Equation 15).

Also, we have
(21)$$\begin{array}{c} \underline{D}=U_{2}\left(\underline{C}\right) \end{array}$$
where
(22)$$\begin{array}{c} \underline{C}=\left(C_{1M},C_{2M},C_{3m},\ldots ,C_{Nm}\right) \end{array}$$ —see (Equation 15) and (Equation 16). In other words, the lower decision level $\underline{D}$ is identified as the loser of the output list $U=U\left(\underline{C}\right)=\left(U_{1}\left(\underline{C}\right),U_{2}\left(\underline{C}\right),\ldots ,U_{N}\left(\underline{C}\right)\right)$ when the currents in $\underline{C}$ are the input. In addition, (Equation 22) shows that the first two currents in $\underline{C}$ are the highest in their class L, while all others currents are the lowest possible in their respective rank. After $\bar{C}$ and $\underline{C}$ are evaluated, they are used as inputs $I_{j}$ in (Equation 2)+(Equation 3) to compute (numerically) the outputs $U\left(\bar{C}\right)$ and $U\left(\underline{C}\right)$. From them, the decision levels $\bar{D}$ and $\underline{D}$ are extracted as in (Equation 19) and (Equation 21), respectively.

Finally, we need the largest $U_{M}$ voltage when applying L. It can be shown that the maximum output voltage $U_{M}$ is obtained if we apply—see [26]–at the input
(23)$$\begin{array}{c} \hat{C}=\left(C_{1M},C_{2m},C_{3m},\ldots ,C_{Nm}\right) \end{array}$$
and take the maximum voltage in the output
(24)$$\begin{array}{c} U_{M}=U_{\sigma 1}\left(\hat{C}\right) \end{array}$$

## 4. Resolutions

In order to appreciate the WTA performances, apart from the threshold $\bar{D}$ and $\underline{D}$, we need a measure of the finesse of selecting the winner. First, we need a measure of the “crowding” of the currents at the input.

The family L contains lists of currents on the $\left[0,I_{M}\right]$ scale, whose cramming is measured by Δ. The difference between the largest and the second-largest current of any list is at least Δ. The coefficient ω defined by
(25)$$\begin{array}{c} \omega =\frac{\Delta }{I_{M}} \end{array}$$

This will be called “THE INPUT RESOLUTION”. When ω is very small, perceiving $I_{w}$ (the winner) and $I_{l}$ (the loser) as distinct from each other is difficult and prone to error. On the output side, the voltages are similarly arranged on the $\left[0,U_{M}\right]$ scale. However, the positions of the *w* (i.e., winner) and *l* (i.e., loser) ranks are now controlled by the decision levels $\bar{D}$ and $\underline{D}$:(26)$$\begin{array}{c} U_{M}\geq U_{w}\geq \bar{D}>\underline{D}\geq U_{l}>0 \end{array}$$
$\bar{D}$ and $\underline{D}$ do not change when a new list from L arrives. Under constraints in (Equation 7)–(Equation 10), $\bar{D}$ and $\underline{D}$ are fixed by (Equation 19) and (Equation 21). Each winner of each list surpasses $\bar{D}$. Each loser of each list in L falls under $\underline{D}$. The gap $\bar{D}-\underline{D}$ compared with the entire $U_{M}$ will be denoted by Ω and called “THE OUTPUT RESOLUTION”:(27)$$\begin{array}{c} \Omega =\frac{\bar{D}-\underline{D}}{U_{M}} \end{array}$$

The similarity between ω at input and Ω at output is complete. Both of them indicate how much of the “reading scale” is taken up by the smallest possible size difference between the *w* and *l* ranks. The circuit is effective if “it amplifies” the resolution of the input list. The large values for Ω/ω mean that the winning rank is highly distinct. To understand the WTA input–output mechanism, we study the function Ω(ω) when $I_{M}$ and $I_{C}$ are unchanged. For clarity, we will translate the results obtained so far in terms of ω, where $\omega =\Delta /I_{M}$.

In [26], it is shown that
(28)$$\begin{array}{c} \Omega \left(\omega \right)>\omega \end{array}$$
at least for a part of ω’s “spectre”. However, examples show that the ratio $\frac{\Omega }{\omega }$ is in the order of hundreds at least.
(29)$$\begin{array}{c} \frac{d\bar{D}\left(\omega \right)}{d\omega }>0 \end{array}$$
(30)$$\begin{array}{c} \frac{d\underline{D}\left(\omega \right)}{d\omega }<0 \end{array}$$
(31)$$\begin{array}{c} \frac{dU_{M}\left(\omega \right)}{d\omega }<0 \end{array}$$

From (Equation 27), it follows immediately that
(32)$$\begin{array}{c} \frac{d\Omega \left(\omega \right)}{d\omega }>0 \end{array}$$
which means that the $\Omega \left(\omega \right)$ function is monotonously increasing. This corresponds to “the intuition” that more disjointed current lists are processed more efficiently (i.e., the gap $\bar{D}-\underline{D}$ is bigger).

## 5. Exploring the Mismatch

The threshold voltage $V_{T}$ is the value of the voltage $V_{GS}$ that controls the transition between the distinct operating regions of the MOS. The value of $V_{T}$ is influenced by the thickness of the oxide layer, as well as by body doping. Also, $V_{T}$ depends a lot on the parasitic charge trapped between oxide and silicon. In contrast to this “accidental” charge, some charge can be introduced intentionally through the process called “ion implantation”.

Moreover, it is well known that subthreshold design has dramatically increased the sensitivity to process variation. This fact is taken into account by introducing the variation in the zero current with threshold voltage. Indeed,
(33)$$\begin{array}{c} I_{0}=I_{0}^{★}\exp\left(-k\frac{V_{T}}{V_{t}}\right) \end{array}$$
where $I_{0}^{★}$ does not depend on $V_{T}$—see [28]. Subsequently, we use (Equation 33) in models (Equation 2) and (Equation 3) and try to evaluate the influence of $V_{T}$ on the output $U_{j}$. Fortunately, we can analytically deduce a qualitative behaviour. Namely, we can show that $U_{j}$ decreases with $V_{T}$:(34)$$\begin{array}{c} \frac{\partial U_{j}}{\partial V_{T}}<0 \end{array}$$

For this, let us denote $F_{j}\left(V,I_{0}\right)=1-\frac{I_{j}}{I_{0}}\exp\left(-k\frac{V}{V_{t}}\right)$ and $d=\exp\left(-\frac{V_{DD}}{V_{t}}\right)$ such that the function in (Equation 4) becomes
(35)$$\begin{array}{c} G\left(V,I_{0}\right)=I_{0}\left[\exp\left(-\frac{V}{V_{t}}\right)-d\right]\sum_{j=1}^{N}F_{j}^{-k}\left(V,I_{0}\right) \end{array}$$

Equation (Equation 3) with a fixed $I_{C}$ gives
(36)$$\begin{array}{c} 0=\frac{\partial G\left(V\left(I_{0}\right),I_{0}\right)}{\partial V}{|}_{I_{0}=cst}\times \frac{\partial V\left(I_{0}\right)}{\partial I_{0}}+\frac{\partial G\left(V\left(I_{0}\right),I_{0}\right)}{\partial I_{0}}{|}_{V=cst} \end{array}$$

We easily obtain
$$\begin{array}{c} \frac{\partial G}{\partial V}\left(V\left(I_{0}\right),I_{0}\right){|}_{I_{0}=cst}<0\;\;\;\mathrm{and}\;\;\;\frac{\partial G}{\partial I_{0}}\left(V\left(I_{0}\right),I_{0}\right){|}_{V=cst}>0 \end{array}$$

Then, (Equation 36) gives
(37)$$\begin{array}{c} \frac{\partial V\left(I_{0}\right)}{\partial I_{0}}>0 \end{array}$$

However, from (Equation 2),
$$\begin{array}{c} U_{j}\left(V\left(I_{0}\right),I_{0}\right)=V_{t}\ln F_{j}\left(V\left(I_{0}\right),I_{0}\right) \end{array}$$
and
$$\begin{array}{c} \frac{\partial U_{j}}{\partial I_{0}}=\frac{\partial U_{j}}{\partial V}{|}_{I_{0}=cst}\times \frac{\partial V\left(I_{0}\right)}{\partial I_{0}}+\frac{\partial U_{j}}{\partial I_{0}}{|}_{V=cst}=V_{t}F_{j}\frac{\partial F_{j}}{\partial V}{|}_{I_{0}=cst}\times \frac{\partial V\left(I_{0}\right)}{\partial I_{0}}+V_{t}I_{j}\frac{\partial F_{j}}{\partial I_{0}}{|}_{V=cst} \end{array}$$

Here, the two derivatives of $F_{j}$ are positive and by also using (Equation 37) we derive $\frac{\partial U_{j}}{\partial I_{0}}>0$. From (Equation 33), we obtain $\frac{\partial U_{j}}{\partial V_{t}}<0$, which is (Equation 34).

## 6. Numerical Checks–Discussion

### 6.1. Decision Levels

In this paragraph, we deal with the decision levels $\bar{D}$ and $\underline{D}$ on the $\left[0,U_{M}\right]$ scale. The known quantities are *k*, $V_{T}$, $I_{0}$ and $V_{t}$ (i.e., the MOS device parameters) then *N*, $I_{C}$ and $V_{DD}$ (circuit parameters) and $I_{M}$ and Δ (i.e., L class characteristics). All these quantities must satisfy the restrictions (Equation 7)–(Equation 10). Their numerical values are k=0.9, $V_{T}=1$ V, $I_{0}=10^{-18}$ Amp, $V_{t}=0.026\;\;$ V, N=100, $I_{C}=10^{-17}$ Amp, $V_{DD}=1.5$ V, $I_{M}=10\;\;$ nA and Δ=0.01 nA. We will call this case “Example 1”. Now follows the analytical part of decision levels calculation. Our result is obtained in formula (Equation 20), (Equation 22) and (Equation 23), which give three particular input lists of currents $\bar{C}$, $\underline{C}$ and $\hat{C}$. For their calculation, we use (Equation 15) and (Equation 16), which lead to $C_{jm}=\left(100-j\right)0.01$ and $C_{jM}=10-\left(j-1\right)0.01$, both in nAmps and for $j\in \bar{1,100}$. Thus, lists $\bar{C}$, $\underline{C}$ and $\hat{C}$ from (Equation 20), (Equation 22) and (Equation 23) are now known. At this point, the numerical part of the calculation begins. We solve the 101 equations in (Equation 2) and (Equation 3) three times corresponding to the inputs $\bar{C}$, $\underline{C}$ and $\hat{C}$. We obtain three output sequence $U\left(\bar{C}\right)$, $U\left(\underline{C}\right)$ and $U\left(\hat{C}\right)$, respectively, each of 100 voltages. From each of them, we select one component according to (Equation 19), (Equation 21) and (Equation 24). Namely, the largest voltage in $U\left(\bar{C}\right)$ is the upper decision level $U_{1}\left(\bar{C}\right)=\bar{D}$. The second-largest voltage in $U\left(\underline{C}\right)$ is the lower decision level $U_{2}\left(\underline{C}\right)=\underline{D}$. Finally, the largest component in $U\left(\hat{C}\right)$ is the maximum possible voltage when any of the L lists is processed: $U_{1}\left(\hat{C}\right)=U_{M}$. For our circuit and device parameters, we obtain $\bar{D}=726$ mV, $\underline{D}=179$ mV and $U_{M}=803$ mV, as shown in Figure 3a. So, out of the scale of 803 mV, any winner will be caught in the interval $\left[\bar{D},U_{M}\right]=\left[726,803\right]$, i.e., in the upper part of 9.6%. The loser will always be in the interval $\left[0,\underline{D}\right]=\left[0,179\right]$, i.e., in the lower part of 22.3%—see Figure 3a. The rest of the scale, i.e., the interval $\left[\underline{D},\bar{D}\right]=\left[179,726\right]$, which represents 68.1% of the total, is the separation “gap” between the unique winner and the rest of the 99 losers.

The ratio $\omega \%=\Delta /I_{M}\%=\frac{0.01}{10}100=0.1\%$ is the “INPUT RESOLUTION” introduced in Section 4. It shows how crowded the lists we intend to process can be. At the output, we brought the “OUTPUT RESOLUTION” $\Omega \%=\left(\bar{D}-\underline{D}\right)/U_{M}\%=\left(726-179\right)/803=68.1\%$—see Figure 3a.

This means that the input resolution yields an input one of 681 times higher. The ratio Ω/ω is a performance index for WTA.

Now, we change $V_{T}$ from 1 V to 1.1 V. For the same parameters, *k*, $I_{0}$, $V_{t}$, *N*, $V_{DD}$, $I_{M}$ and Δ, we obtain the lists $\bar{C}$, $\underline{C}$ and $\hat{C}$. Solving again the 101 equations from (Equation 2) and (Equation 3) (this time with $V_{T}=1.1$ V), we obtain $U_{1}\left(\bar{C}\right)=\bar{D}=515$ mV, $U_{2}\left(\underline{C}\right)=\underline{D}=179\;\;$ mV and $U_{1}\left(\hat{C}\right)=U_{M}=589$ mV. Then, Ω%=57%—see Figure 3b and the comments in Section 6.2.

### 6.2. Mismatch

We take the device parameters in Example 1 except for $I_{0}$, which is replaced by (Equation 33) with $I_{0}^{★}=10^{-33}$ Amp. Also, we successively use in (Equation 33) $V_{T}$ as 1 V, 1.02 V, 1.04 V, *…*, 1.1 V. For the class L with $I_{M}=10$ nA and Δ=0.01 nA, N=100, we follow the procedure in Section 6.1 and determine $U_{M}$, $\bar{D}$ and $\underline{D}$ for each of these six cases. The results are presented in Table 1.

We notice that major effects occur when the threshold voltage $V_{T}$ increases by 10%—See Figure 3a,b. The maximum voltage $U_{M}$ decreases drastically by 27%, while the higher decision level $\bar{D}$ decreases by 29%. Remarkably, the lower decision level $\underline{D}$ is hardly influenced by the deviation of $V_{T}$. Even more remarkable is that the decrease in the output resolution by 10 percent does not sufficiently reflect the major worsening of the accuracy in the appreciation of the maximum $U_{\sigma 1}$. It turns out that the maximum output voltage $U_{M}$ must accompany the output resolution parameter Ω in the WTA circuit specifications.

### 6.3. List Processing

Example 3

With the WTA data from the previous example, let us take a list of 100 currents given by
(38)$$\begin{array}{c} I_{2j}=\left(2j+1\right)\Delta ,\;\;\;j\in \bar{1,24} \end{array}$$
(39)$$\begin{array}{c} I_{50+2j}=\left(197-4j\right)\Delta ,\;\;\;j\in \bar{0,25} \end{array}$$
(40)$$\begin{array}{c} I_{2j+1}=4j\Delta ,\;\;\;j\in \bar{0,49} \end{array}$$

Among these 100 currents two groups are shown in Table 2 Column 1. The largest current in our list $I=\left(I_{1},I_{2},\ldots ,I_{100}\right)$ is found at terminal 50. If $\sigma =\left(\sigma_{1},\sigma_{2},\ldots ,\sigma_{100}\right)$ is the permutation that gives the descending order, then $\sigma_{1}=50$, i.e., $I_{\sigma_{1}}=I_{50}=1.97$ nA, as in Table 2 Column 1. Also, the second-largest current is at terminal 99. Thus, $\sigma_{2}=99$, i.e., $I_{\sigma_{2}}=I_{99}=1.96$ nA, as in Table 2. Now, we solve the Equations (Equation 2) and (Equation 3) with the above currents and $V_{T}=1$ V. Out of the output $U=\left(U_{1},U_{2},\ldots ,U_{100}\right)$, Table 2 Column 2 shows the voltages $U_{49}$ to $U_{54}$ and $U_{95}$ to $U_{100}$. It is verified that the order in *U* is given by the same permutation σ as currents, such that the winner is $U_{\sigma 1}=U_{50}=749$ mV and the loser is $U_{\sigma 2}=U_{99}=137$ mV—see Table 2 Figure 3. The winner is caught in the interval $\left[\bar{D},U_{M}\right]$, while the loser $U_{\sigma 2}$ together with all other voltages $U_{\sigma 3},\ldots ,U_{\sigma 100}$ fall in the interval $\left[0,\underline{D}\right]$. The output resolution is Ω%=68%, way better than ω%=0.1% at the input.

Next, we change the threshold voltage $V_{T}$ (Table 2 Column 3) to 1.1 V and obtain an output similarly ordered, i.e., the winner is $U_{50}$ and the loser is $U_{99}$. Their placement above $\bar{D}=515$ mV and under $\underline{D}=179$ mV, respectively, (see Table 1) shows the correctitude of detaching the largest element of the input list.

## 7. Conclusions

Finding the maximum in continuous signal strings is a fundamental operation in signal processing. When processing speed is essential, the analog version is preferable. In this framework, the WTA circuit has imposed itself through technological simplicity. Of course, in this case we have to solve the problem of precision in separating the maximum rank (“winner”) from the next rank (“loser”). This paper, in close connection with [26], verifies numerically that “the decision levels” work correctly. For this, the theoretical notions of decision levels, resolution and mismatch are specified first. Then, a class of strings of 100 currents is considered for which the decision levels and input and output resolution are analytically calculated. Numerical processing follows that simulates the operation of the WTA circuit. An ordered string of voltages is obtained at the output. It is verified that the largest element of this string exceeds the upper decision level while the rest of the elements are crowded much further, namely, below the lower decision level. The output resolution compared to the input resolution indicates how much “the winner” is separated, i.e., the effectiveness of the WTA. It is theoretically shown that any of the output voltages decreases monotonically with increases in $V_{T}$. It is then verified by numerical calculation that the small increase in the threshold voltage of the MOS transistors leads to a drastic decrease in both the decision levels and the output resolution. Since the manufacturing technology cannot ensure a really constant $V_{T}$ for series production, this is a major problem in the design of MOS circuits, especially those that work in the subthreshold.

## Figures and Tables

**Figure 1 sensors-23-06247-f001:**
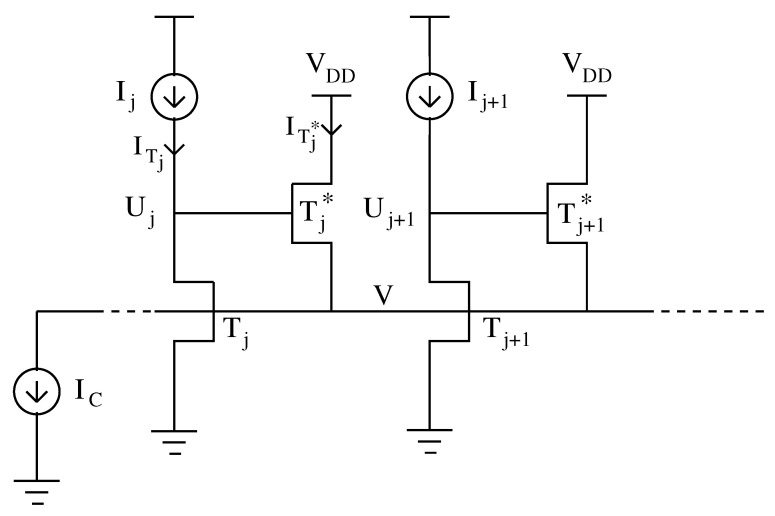
Two cells of Lazzaro WTA. $I_{1},I_{2},\ldots ,I_{N}$ are input currents; $U_{1},U_{2},\ldots ,U_{N}$ are output voltages; $I_{C}$ is the bias current.

**Figure 2 sensors-23-06247-f002:**
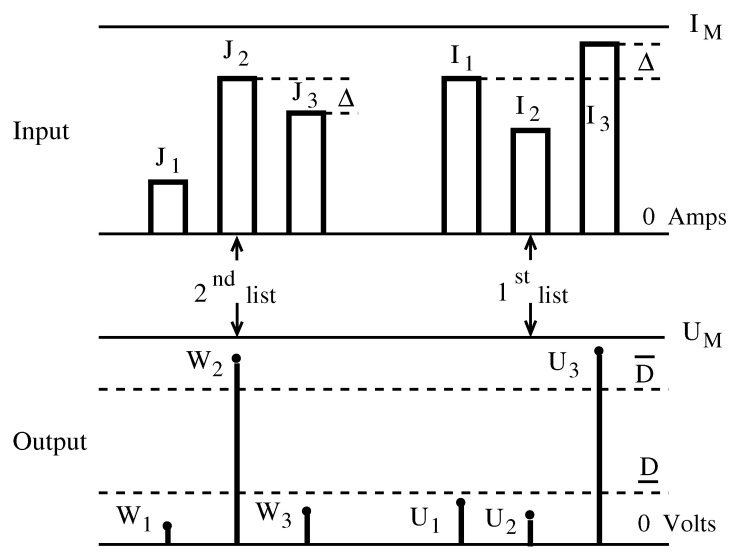
The input list $\left(I_{1},I_{2},I_{3}\right)$ yields the output list $\left(U_{1},U_{2},U_{3}\right)$; the input list $\left(J_{1},J_{2},J_{3}\right)$ yields the output list $\left(W_{1},W_{2},W_{3}\right)$. The winning ranks are “3” in the first case and “2” in the second, since $U_{3}$ and $W_{2}$ surpass $\bar{D}$.

**Figure 3 sensors-23-06247-f003:**
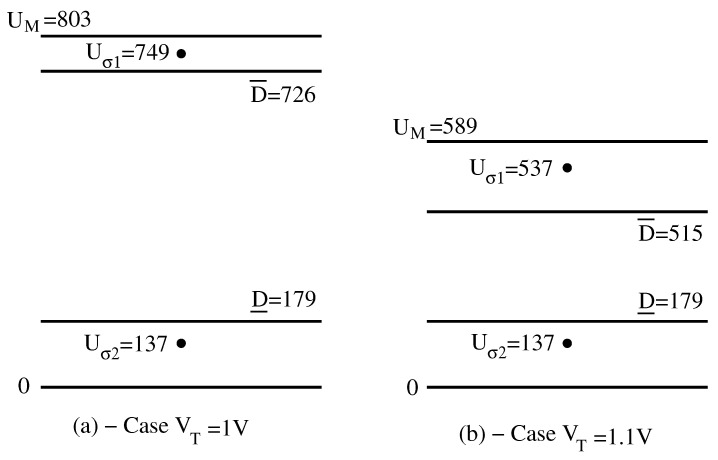
Example 1: N=100, $I_{M}=10\;\;mAmp$ and Δ=0.01 nAmp. Two cases for $V_{T}=1$ V and $V_{T}=1.1$ V. The winner $U_{\sigma 1}$, the loser $U_{\sigma 2}$, the maximum voltage $U_{M}$ and the decision levels $\bar{D}$ and $\underline{D}$.

**Table 1 sensors-23-06247-t001:** Example 2 Section 6.2. $V_{T}$, $U_{M}$, $\bar{D}$ and $\underline{D}$ in mV. Output resolution Ω in percent.

$V_{T}$	1000	1020	1040	1060	1080	1100
$U_{M}$	803	759	719	674	632	589
$\bar{D}$	726	684	642	600	557	515
$\underline{D}$	179	179	179	179	179	179
$\frac{\bar{D}-\underline{D}}{U_{M}}\;\;\%$	68%	66%	64%	62%	59%	57%

**Table 2 sensors-23-06247-t002:** Examples 3 and 4. $I_{49}-I_{54}$ and $I_{95}-I_{100}$, two groups of currents given in (Equation 38)–(Equation 40) are listed in Column 1. Column 2 shows output voltages for $V_{T}=1$ V. Column 3 shows output voltages for $V_{T}=1.1$ V. Column 4 shows the indices σj—the *j*-th current (and voltage) in descending order.

	Example 3	Example 4	
$I_{j}$ nA	$U_{j}\;\;\mathrm{in}$ mV	$U_{j}\;\;\mathrm{in}$ mV	$\sigma_{j}$
	$V_{T}=1$ V	$V_{T}=1.1$ V	
$I_{49}=0.96$	$U_{49}=17.5$	$U_{49}=17.3$	$\sigma_{24}=49$
$I_{50}=1.97$	$U_{50}=749$	$U_{50}=537$	$\sigma_{1}=50$
$I_{51}=1$	$U_{51}=18$	$U_{51}=18$	$\sigma_{50}=51$
$I_{52}=1.93$	$U_{52}=101$	$U_{52}=101$	$\sigma_{3}=52$
$I_{53}=1.04$	$U_{53}=19$	$U_{53}=19.5$	$\sigma_{48}=53$
$I_{54}=1.89$	$U_{54}=83$	$U_{54}=83$	$\sigma_{5}=54$
$I_{95}=1.88$	$U_{95}=80$	$U_{95}=80$	$\sigma_{6}=95$
$I_{96}=1.05$	$U_{96}=19.5$	$U_{96}=19$	$\sigma_{47}=96$
$I_{97}=1.92$	$U_{97}=95$	$U_{97}=95.5$	$\sigma_{4}=97$
$I_{98}=1.01$	$U_{98}=18.5$	$U_{98}=18$	$\sigma_{49}=98$
$I_{99}=1.96$	$U_{99}=137$	$U_{99}=137$	$\sigma_{2}=99$
$I_{100}=0.97$	$U_{100}=17$	$U_{100}=17$	$\sigma_{51}=100$

## Data Availability

All of the relevant research data will be made available upon request after the publication of the paper.

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
