# Peer review of "Decision Levels and Resolution for Low-Power Winner-Take-All Circuit†"

_sensors, 2023, doi:10.3390/s23146247_

Round 1

Author Response

4 July 2023

To: Sensors Editorial Office, Professor Erin Hu and Ms. Eliza Wu

From R. L. Costea

Revision of “DECISION LEVELS AND RESOLUTION FOR LOW POWER WINNER TAKE ALL CIRCUIT”

Dear Editor,

Find below a major revision of my paper. This consists of:

  1. Complete rewriting of chapter 5 examples.
  2. Changing the order of examples 2 and 3.
  3. Rewriting of Section 6 Conclusion.

    REFERENT 1

    Question 1

    Is it possible to include some numerical results in graphical manner in order demonstrate the improvement?

    Answer 1:

    You are probably right. But even if I was making a graph, I couldn’t give up the tables and the text was getting too long.

    With regards,

    Ruxandra Costea

Reviewer 2 Report

DECISION LEVELS AND RESOLUTION FOR LOW POWER WINNER TAKE ALL CIRCUIT

The article is well-written and appears to be up-to-date with the current state of the art. However, there are some grammar and typographical errors that need to be corrected.

It would be helpful to ensure that all tables and equations are listed properly whenever necessary.

Comments/Queries:

How does the paper acknowledge the importance of Winner Take All (WTA) circuits in applications where the selection of the largest element in a sequence of currents is required?

What is the specific focus of the paper when it comes to WTA circuits? How does it contribute to our understanding of their design, operation, and performance characteristics?

Why is it significant to explore the operation of the WTA Lazzaro circuit in the subthreshold regime? What advantages does operating in this region offer?

How does the paper introduce and define the concept of "decision levels" in the WTA circuit? What is the significance of this recent development?

What is the purpose of the numerical verification and discussion conducted in the paper? How does it contribute to our understanding of the behavior and performance of the WTA circuit?

In what ways does the investigation of decision levels and their dependence on various parameters in the paper provide valuable information for circuit and device optimization? How can researchers and engineers utilize this information?

Although not explicitly mentioned, what are some potential applications of WTA circuits in various fields? How do the findings and insights from this paper contribute to these potential applications?

Regarding Table 1:

What information does Table 1 provide regarding the two groups of currents, I49-I54 and I95-I100?

How are the output voltages in Columns 2 and 3 of Table 1 related to the currents listed in Column 1?

What does the threshold voltage represent in Table 1, and how does it affect the output voltages?

What is the significance of the values of σj listed in Column 4 of Table 1?

How does the order of the currents in descending order contribute to our understanding of the behavior of the WTA Lazzaro circuit?

How does Table 1 illustrate the relationship between the currents, output voltages, and threshold voltages in the WTA Lazzaro circuit?

How can the information presented in Table 1 be utilized to analyze the performance and characteristics of the WTA Lazzaro circuit?

Regarding Table 2:

What information does Table 2 provide regarding Example 3?

How are the threshold voltage (V_T), maximum voltage (U_M), decision levels (D), and output resolution related in the context of Example 3?

Can you explain the significance of the output resolution presented in Table 2?

Are there multiple values of V_T presented in Table 2? If so, what are they, and how do they affect the other parameters?

How can the information in Table 2 be used to analyze or evaluate the performance of the system or circuit considered in Example 3?

Based on the given texts, what potential implications or insights can be drawn from the values presented in Table 2?

Moderate editing of English language required

Author Response

4 July 2023

To: Sensors Editorial Office, Professor Erin Hu and Ms. Eliza Wu

From R. L. Costea

Revision of “DECISION LEVELS AND RESOLUTION FOR LOW POWER WINNER TAKE ALL CIRCUIT”

Dear Editor,

Find below a major revision of my paper. This consists of:

  1. Complete rewriting of chapter 5 examples.
  2. Changing the order of examples 2 and 3.
  3. Rewriting of Section 6 Conclusion.

Below you can find the answers to the 20 questions raised by Referent 1.

REFERENT 1

Question1:

How does the paper acknowledge the importance of Winner Take All (WTA) circuits in applications where the selection of the largest element in a sequence of currents is required?

Answer1:

By its subject and by its goal the paper belongs to a recent trend of analog processing. For small signals it competes very well with its digital counterpart.

Question 2:

What is the specific focus of the paper when it comes to WTA circuits? How does it contribute to our understanding of their design, operation, and performance characteristics?

Answer 2:

Three area of interest is enhancement of accuracy for WTA.

Question 3:

Why is it significant to explore the operation of the WTA Lazzaro circuit in the subthreshold regime? What advantages does operating in this region offer?

Answer 3:

The subthreshold regime for MOS devices is used when WTA works with nanoamp signals. It has the excellent exponential dependence of drain to source current by the gate to source voltage.

Question 4:

How does the paper introduce and define the concept of "decision levels" in the WTA circuit? What is the significance of this recent development?

Answer 4:

“Decision levels” are concept recently introduced and published in CSSP 2022, “Marinov, C.A., Costea, R.L. Designing a Winner–Loser Gap for WTA in Subthreshold. Resolution Performance Revisited. Circuits Syst Signal Process 41, 7145–7171 (2022). https://doi.org/10.1007/s00034-022-02100-9”, by the author. In the present paper we explain the notion (see (19)-(24)) and used it for subsequent theory and numerical examples.

Question 5:

What is the purpose of the numerical verification and discussion conducted in the paper? How does it contribute to our understanding of the behaviour and performance of the WTA circuit?

Answer 5:

Since the decision levels is intended to isolate the largest element in a list, it is natural to verify if this happens.

Question 6:

In what ways does the investigation of decision levels and their dependence on various parameters in the paper provide valuable information for circuit and device optimization? How can researchers and engineers utilize this information?

Answer 6:

The author hopes that “Decision levels” and their derivation “INPUT” and ”OUTPUT” resolution are valuable for improving the accuracy of WTA design.

Question 7:

Although not explicitly mentioned, what are some potential applications of WTA circuits in various fields? How do the findings and insights from this paper contribute to these potential applications?

Answer 7:

The WTA in subthreshold has large applications in processing signals originating in sounds, light, odor, movement of objects, computer memory, neuromorphic circuits, prosthetics. Our paper tries to enhance WTA precision.

Regarding Table 1

Question 8:

What information does Table 1 provide regarding the two groups of currents, I49-I54 and I95-I100?

Answer 8:

The two groups of currents belong to the 100 currents list of input.

Question 9:

How are the output voltages in Columns 2 and 3 of Table 1 related to the currents listed in Column 1?

Answer 9:

The voltages in column 2 represent two groups from the 100 voltages yield at output in case VT=1V. Similarly, for column 3 in case VT=1.1V.

Question 10:

What does the threshold voltage represent in Table 1, and how does it affect the output voltages?

Answer 10:

The voltage VT is a MOS transistor parameter which in our case separates the weak inversion.

Question 11:

What is the significance of the values of σj listed in Column 4 of Table 1?

Answer 11:

σj is the natural index (out of 1,2,…100) indicating the rank of input current Iσj out of the input list. It translates to the output voltages Uσj which is the rank.

Question 12:

How does the order of the currents in descending order contribute to our understanding of the behavior of the WTA Lazzaro circuit?

Answer 12:

The rank σ1 is the largest current of input list. We call it “the winner” as distinct from “the loser” rank σ2 which marks out the second largest current. The rest of indices σ3, σ4,…., σ100 denote the corresponding size order of currents (voltages).

Question 13:

How does Table 1 illustrate the relationship between the currents, output voltages, and threshold voltages in the WTA Lazzaro circuit?

Answer 13:

Table 1 shows the rank of each current (voltage) among the 100 components.

Question 14:

How can the information presented in Table 1 be utilized to analyze the performance and characteristics of the WTA Lazzaro circuit?

Answer 14:

The WTA increases the gap between σ1 and σ2 ranks from output relative to input.

Regarding Table 2:

Question 15:

What information does Table 2 provide regarding Example 3?

Answer 15:

Table 2 brings together the performance indices U_M, overline D, underline D, Omega of our WTA for 6 different V_T.

Question 16:

How are the threshold voltage (V_T), maximum voltage (U_M), decision levels (D), and output resolution related in the context of Example 3?

Answer 16:

These are performance indices of WTA defined in Section 2 and 3.

Question 17:

Can you explain the significance of the output resolution presented in Table 2?

Answer 17:

As explained in Section 3, the output resolution Ω = overlineD −underline D /U_M measures the minimum distance U_sigma1 to U_sigma2 relative to the [0,U_M] scale. It is specific a class of infinite number of input lists characterized by I_M and Delta.

Question 18:

Are there multiple values of V_T presented in Table 2? If so, what are they, and how do they affect the other parameters?

Answer 18:

Yes, there are 6 different V_T values. V_T MOS parameter is essential for mismatch property of circuit.

Question 19:

How can the information in Table 2 be used to analyze or evaluate the performance of the system or circuit considered in Example 3?

Answer 19:

All items in Table 2 (i.e., overlineD, underline D, U_M, Omega) are proposed for evaluation of accuracy of WTA. They depend essentially on the threshold voltage V_T.

Question 20:

Based on the given texts, what potential implications or insights can be drawn from the values presented in Table 2?

Answer 20:

Table 2 shows that the smaller V_T is the larger output resolution is. And the growth slope is very high. Unfortunately, technological control cannot ensure extra-fine adjustment of V_T.

With regards,

Ruxandra Costea

Round 2

Reviewer 2 Report

Good Job

Minor editing of English language required